# Determination of Water Quality Characteristics and Nutrient Exchange Flux at the Sediment—Water Interface of the Yitong River in Changchun City, China

**Ke Zhao** [1], **Hang Fu** [1], **Qian Wang** [1] and **Hai Lu** [2,*]

[1]    Key Laboratory of Songliao Aquatic Environment, Ministry of Education, Jilin Jianzhu University, Changchun 130118, China; zhaoke326@126.com (K.Z.); fuhang8652@163.com (H.F.); w9688745@126.com (Q.W.)

[2]    College of Civil Engineering and Architecture, Changchun Sci-Tech University, Changchun 130022, China

*    Correspondence: haimm110@126.com; Tel.: +86-431-84566151

**Abstract:** In this paper, the characteristics of water pollution in Yitong River were analyzed by the comprehensive pollution index method. Combined with the pore water concentration gradient method and Fick's first law, the release characteristics of nutrients at the sediment–water interface of Yitong River (Jilin Province, China) were studied. The results showed that the distribution trend of nitrogen and phosphorus content in the overlying and interstitial water of the Yitong River was the same, and the highest values appeared at the S3 and S5 points in the urban section. The water quality was mainly affected by nitrogen pollutants in domestic sewage. The evaluation results of the water quality comprehensive pollution index showed that the pollution degree of interstitial water in urban areas was much higher than that of the overlying water, and the endogenous nitrogen and phosphorus pollutants had the risk of diffusion to the overlying water. The exchange flux analysis of ammonia nitrogen ($NH_4^+$-N), total dissolved nitrogen (TDN), and total dissolved phosphorus (TDP) in water showed that the diffusion flux of $NH_4^+$-N ranged from 0.03 to 6.52 mg·(m$^2$·d)$^{-1}$, and the sediment was the "source" of ammonia nitrogen pollutants. The range of TDN diffusion flux was $-1.57$ to 11.6 mg·(m$^2$·d)$^{-1}$, and the difference between points was large. The sediment was both the "source" and "sink" of nitrogen pollutants. The range of TDP diffusion flux was $-0.05$ to 0.22 mg·(m$^2$·d)$^{-1}$. Except for point S8, the TDP diffused from sediment into the water body. Among all the sampling points, the diffusion fluxes of $NH_4^+$-N, TDN, and TDP at the S3 point were the largest, the release rate of endogenous pollutants was the most rapid, and the pollution to the water quality was the most serious. The results are of great significance to the exchange flux of nutrients at the sediment–water interface of rivers and the prevention and control of water eutrophication. It also provides a reference for the study of nutrient exchange flux at the sediment–water interface of rivers and other surface water bodies worldwide.

**Keywords:** Yitong River; overlying water; interstitial water; water quality characteristics; nitrogen and phosphorus nutrients; diffusion flux

## 1. Introduction

In recent years, with the rapid development of the economy and the continuous improvement of living standards, the problem of river eutrophication in China has become increasingly prominent, which seriously threatens the water environment quality and water safety [1]. Nitrogen and phosphorus are the most critical nutrients affecting water quality and are usually the main limiting factor of water eutrophication [2–4]. The migration and transformation of nitrogen and phosphorus at the sediment–water interface play an essential role in the aquatic ecological environment [5,6]. River sediment is a vital storage place of nitrogen and phosphorus nutrients in water bodies. When the water environment is disturbed, the nutrient substances with high content in sediment are released to the

overlying water by diffusion, convection, and resuspension [7–9]. Furthermore, this phenomenon mainly results in the content of nitrogen and phosphorus exceeding the standard limits.

When Kuwabara [10] studied the source of phosphorus nutrients in Coeur d'Alene Lake in the United States, it was found that the release of endogenous phosphorus was equivalent to the input of exogenous phosphorus. The study by Berelson [11] on the nutrient exchange in Port Phillip Bay found that the release of endogenous nitrogen and phosphorus nutrients accounted for more than 50% of the total source of nutrients. Therefore, river management should control the input of exogenous nitrogen and phosphorus pollutants and effectively control the release of endogenous pollutants in sediments [12–14]. At the same time, it is of great significance to study the distribution characteristics of nutrients and the interfacial exchange flux of sediment, interstitial water, and overlying water, and to analyze the present situation and the source of nitrogen and phosphorus pollution in water bodies [15,16]. In addition, the release risk of nitrogen and phosphorus at the sediment–water interface is closely related to the mass concentration, form, and organic matter content of nitrogen and phosphorus in overlying water [17,18]. At present, many scholars have studied the content, occurrence form, temporal and spatial distribution, pollution evaluation, and release mechanism of nutrients in sediments of rivers and lakes [19–22]. While the research methods of sediment nutrient release flux mainly include the pore water concentration gradient method [23], laboratory culture method [24], and in situ box determination method [25], the pore water concentration gradient method is widely used because it can fit the actual situation and has less workload.

The Yitong River is the mother river of Changchun City, running through the city from the south to the north, is the most crucial source of water for industrial water and irrigation water, and its upstream Xinlicheng reservoir is the drinking water source of Changchun City. In recent years, the research on the Yitong River mainly focuses on water pollution, sediment pollution, and biological community, while its endogenous nutrient load is not clear [26,27]. Thus, this paper takes the Yitong River as a case study and analyses nitrogen and phosphorus's distribution characteristics and exchange flux between sediment and water interface. The effect of nitrogen and phosphorus release on water quality was evaluated by determining whether the sediment was the source or sink of nitrogen and phosphorus nutrients. The results can provide reference for the precise remediation of nitrogen and phosphorus pollution in the Yitong River and provide data accumulation for the distribution characteristics of nutrients in the water and sediment and the nutrient exchange flux at the sediment–water interface of North China rivers. The study also provides a scientific basis for the source analysis and eutrophication treatment of rivers and other surface water worldwide, with significant theoretical and application value.

## 2. Materials and Methods

### 2.1. Overview of the Study Area and Sampling Points

The total length of the Yitong River is 342.5 km and flows through 5 counties and cities, including Panshi, Yitong, Changchun, Nong'an, and Dehui. As the secondary tributary of the Songhua River, the water environment of the Yitong River has a direct impact on the water quality safety of the Songhua River. In recent years, comprehensive treatment of the Yitong River has been carried out in Changchun City, and the water quality has been significantly improved. However, among the five national and provincial assessment sections of the Yitong River, the water bodies of Yangjiaweizi, Baolong Bridge, and Kaoshan Bridge have severe nitrogen and phosphorus pollution. Furthermore, site sampling shows that most of the sediments in the Yitong River are silt, and the sediments in the urban section are in a black and smelly state, with more pollutants and severe endogenous pollution. Therefore, this study took the Yitong River as the study area; the overlying water and sediment samples of the Yitong River were collected in August 2021. The sampling points were selected from the five points on the national and provincial control sections (Xingguang section (S1), under the dam of Xinlicheng Reservoir (S2), Yangjiaweizi (S5),

Baolong Bridge section (S7), and Kaoshan Bridge section (S8)). In addition to the 5 points, according to the characteristics of the water quality, three points were selected in front of the south gate of the South Third Ring Road (S3), in front of the Free Gate (S4), and under the Beihu Bridge (S6). Therefore, the total number of sampling points was 8, and the layout of them is shown in Figure 1.

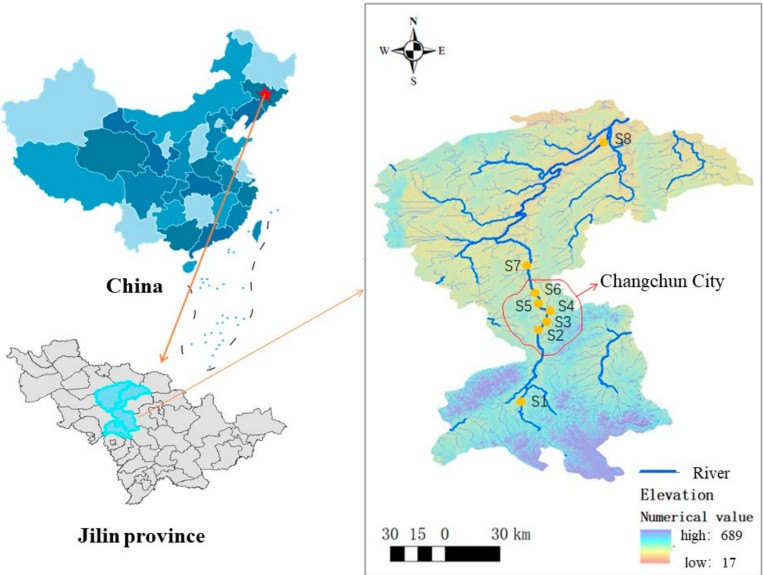

**Figure 1.** Schematic diagram of the distribution of water samples and sediment sampling points of the Yitong River.

*2.2. Sample Collection and Test Method*

The overlying water was collected using the stainless-steel water collector into a polyethene sampling bottle. The sediment samples of 0–10 cm under the surface of the river's bottom were collected with a Peterson dredger and sealed in a polythene bag. The water and sediment samples were transported back to the laboratory at low temperatures throughout the process. One part of sediment samples was centrifuged to collect interstitial water, and the other was dried naturally. After grinding, they were screened through 100 mesh sieves for storage.

At the same time of sampling, the dissolved oxygen (DO), power of hydrogen (pH), water temperature (WT), and water depth of the water body were measured. The overlying and interstitial water were filtered by a glass fiber filter membrane with an aperture of 0.45 μm, and then the nutrient indexes were determined. The determination methods were carried out according to Method of Water and Wastewater Monitoring (Fourth Edition) issued by the Ministry of environmental protection of the people's Republic of China [28]. All water quality indexes were determined within 24 h. The determination of nitrogen and phosphorus in sediments referred to Methods of Soil Agrochemical Analysis [29]. The specific physical and chemical indexes and their detection methods in the water and sediment determined in this experiment are shown in Table 1.

**Table 1.** Determination Methods of conventional indexes in water and sediments [28,29].

| General Indicators | Determination Method |
|---|---|
| pH | Portable pH meter |
| DO | Portable DO instrument |
| $COD_{Mn}$ | Acid potassium permanganate method |
| TN (in water) | Potassium persulfate oxidation–ultraviolet spectrophotometry |
| TP (in water) | Molybdenum antimony resistance Spectrophotometry |
| $NH_4^+$-N | Nessler reagent spectrophotometry |
| TN (in sediment) | Kelvin method |
| TP (in sediment) | Sodium carbonate melting method |

*2.3. Calculation and Evaluation Method*

2.3.1. Water Quality Evaluation Method

The Comprehensive pollution index of water quality was calculated based on a single pollution index evaluation. All or representative pollutants were selected to participate in the assessment according to the needs when selecting indicators. In this paper, considering the pollution characteristics of the Yitong River, representative pollutants were selected when calculating the comprehensive pollution index of water quality, including WT, pH, DO, $COD_{Mn}$ (permanganate index), $NH_4^+$-N, TN (total nitrogen), and TP (total phosphorus). The river pollution degree can be classified according to the pollution index. The calculation method is as follows [30]:

$$P = \frac{1}{n}\sum_{i=1}^{n} P_i, \; P_i = \frac{C_i}{C_0} \tag{1}$$

where P is the comprehensive pollution index, n is the number of water quality factors involved in the evaluation, $P_i$ is the pollution index of a water quality factor, $C_i$ is the measured concentration of a water quality factor ($mg \cdot L^{-1}$), $C_0$ is the Class III standard limit ($mg \cdot L^{-1}$) of a water quality factor in the Environmental Quality Standard for Surface Water (GB 3838-2002). The standard limits of fundamental indicators are shown in Table 2.

**Table 2.** Standard limits of fundamental indicators of surface water environmental quality standards ($mg \cdot L^{-1}$) [31].

| Index | Class I | Class II | Class III | Class IV | Class V |
|-------|---------|----------|-----------|----------|---------|
| WT | | | None | | |
| pH | | | 6–9 | | |
| DO$\geq$ | 7.5 | 6 | 5 | 3 | 2 |
| $COD_{Mn}\leq$ | 2 | 4 | 6 | 10 | 15 |
| $NH_4^+$-N$\leq$ | 0.15 | 0.5 | 1.0 | 1.5 | 2.0 |
| TP$\leq$ | 0.02 | 0.1 | 0.2 | 0.3 | 0.4 |
| TN$\leq$ | 0.2 | 0.5 | 1.0 | 1.5 | 2.0 |

When $P \leq 0.20$, the water quality grade is good; $0.20 < P \leq 0.40$, the water quality grade is better; $0.40 < P \leq 0.70$, the water quality level is slightly polluted; $0.70 < P \leq 1.00$, the water quality is moderately polluted; $1.00 < P \leq 2.00$, the water quality level is severely polluted; and $P > 2.00$, the water quality is seriously polluted.

2.3.2. Calculation Method of Nitrogen and Phosphorus Diffusion Flux in Sediments

The exchange of nitrogen and phosphorus nutrients at the sediment–water interface is mainly realized by molecular diffusion caused by a concentration difference. In this study, Fick's first law was used to calculate the net fluxes of ammonia nitrogen, dissolved total nitrogen, and dissolved total phosphorus. The calculation formulas are as follows [32,33]:

$$F = \varphi \times D_s \times \frac{\partial_c}{\partial_x} \tag{2}$$

where F is the diffusion flux of a substance at the sediment–water interface ($mg \cdot m^{-1} \cdot d^{-1}$); $\varphi$ is sediment porosity; $\frac{\partial_c}{\partial_x}$ is the concentration gradient of nutrients at the sediment–water interface ($mg \cdot m^{-4}$); and $D_S$ is the diffusion coefficient of nutrients at the sediment–water interface ($cm^2 \cdot s^{-1}$). $D_S$ includes the bending effect of sediments, but it is not easy to measure the bending of sediments in practical research, so the porosity of sediments $\varphi$ and diffusion coefficient $D_0$ of the ideal solution are often used to deduce it [34], and the relationship is as follows:

$$D_s = \varphi D_0 \; (\varphi < 0.7) \tag{3}$$

$$D_s = \varphi^2 D_0 \; (\varphi \geq 0.7) \tag{4}$$

where $D_0$ is the diffusion coefficient of the ideal solution; the Ideal diffusion coefficient of $NH_4^+$-N is $D_0 = 9.8 \times 10^{-6}$ cm$^2 \cdot$s$^{-1}$; Ideal diffusion coefficient of TDN is $D_0 = 14.21 \times 10^{-6}$ cm$^2 \cdot$s$^{-1}$; and Ideal diffusion coefficient of TDP is $D_0 = 6.12 \times 10^{-6}$ cm$^2 \cdot$s$^{-1}$. Porosity is calculated as follows [35]:

$$\varphi = \frac{(W_w - W_d) \times 100\%}{(W_w - W_d) + \frac{W_d}{\rho}} \tag{5}$$

where $W_w$ and $W_d$ represents the fresh weight and dry weight of sediment (g), respectively, $\rho$ represents the ratio of the average density of sediment to water density, usually 2.5.

### 2.3.3. Calculation of Contribution Rate of Endogenous Nutrient Release to the Overlying Water Body

If only molecular diffusion is considered as the primary way for the migration and transformation of nutrients at the interface, the ratio of nutrients in sediment interstitial water to overlying water can represent the impact degree of interface nutrient diffusion on water quality, and the calculation formula is as follows [36]:

$$a = \frac{F \times T_w / h}{c} \tag{6}$$

where $a$ represents the contribution rate of nutrient diffusion to the overlying water body (%); $F$ is the release flux of nutrients at the sediment–water interface (mg$\cdot$m$^{-1} \cdot$d$^{-1}$); $T_w$ represents the retention time of water body (d); $h$ is the river water depth; and $c$ represents the concentration of a nutrient in the overlying water (mg$\cdot$L$^{-1}$).

### 2.3.4. Data Analysis

The schematic diagram of the distribution of water samples and sediment sampling points of the Yitong River was drawn by ArcGIS10.5 (ESRI, RedLands, CA, USA), the data graphs were drawn by Origin9.0 (OriginLab, Northampton, MA, USA) software, the data correlation analysis was obtained by Pearson correlation analysis in SPSS20.0 (IBM, New York, NY, USA) software.

## 3. Results and Discussion

### 3.1. Analysis of Water Quality Characteristics of Overlying Water and Interstitial Water

#### 3.1.1. Analysis of Water Quality Characteristics of Overlying Water

The concentrations of $NH_4^+$-N, TDN, and TDP in the overlying water of the Yitong River are shown in Figure 2. It can be seen from Figure 2 that the concentration of $NH_4^+$-N in the overlying water was between 0.11 and 2.87 mg$\cdot$L$^{-1}$. Among them, the concentration of $NH_4^+$-N in the two adjacent residential areas of urban section S3 and S5 was high, while the $NH_4^+$-N values of other points were within the standard limit of Class II water body of surface water ($\leq$0.5 mg$\cdot$L$^{-1}$). This showed that the high concentration of ammonia nitrogen produced by the decomposition of organic pollutants in domestic sewage by denitrifying bacteria was the primary source of ammonia nitrogen in the overlying water body of the river in summer. The TDN concentration in the overlying water was between 0.38 and 7.8 mg$\cdot$L$^{-1}$ and the highest value still occurred at S3 and S5 points in the urban section, where the TDN concentration exceeded 7 mg$\cdot$L$^{-1}$. This may be due to the severe nitrogen pollution caused by the significant content of nitrogen pollutants in the middle reach of the river flowing downstream. The concentration of TDP was between 0.027 and 0.59 mg$\cdot$L$^{-1}$, the concentration of S3 and S5 in the urban section exceeded 0.5 mg$\cdot$L$^{-1}$. The content of phosphorus pollutants in S8 also seriously exceeded the standard value.

In general, the water quality in the upper reaches of the Yitong River was good, and all water quality indexes of S2 did not exceed the Class II of the surface water standard ($\leq$0.5 mg$\cdot$L$^{-1}$). However, the water quality in the urban section of the middle reaches was poor, and the nitrogen and phosphorus pollution at S3 and S5 were the most serious, mainly due to the increase in human activities, as well as the increase in domestic sewage in

summer, resulting in the deterioration of water quality. In addition, the inflow of pollutants in the middle reaches and the direct discharge of domestic sewage and waste into the river affected the downstream water quality, resulting in severe water pollution.

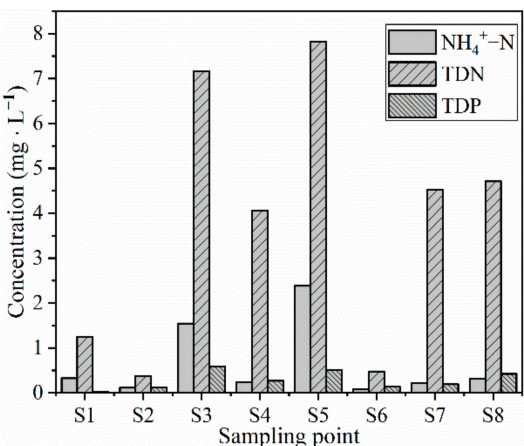

**Figure 2.** Nutrient concentration of overlying water at sampling point of the Yitong River.

The $COD_{Mn}$, DO, and pH values of overlying water of the Yitong River are shown in Table 3. The variation range of dissolved oxygen in each point of the Yitong River was 4.1–9.0 mg·L$^{-1}$. The measured values at all sampling points were widely dispersed, and the dissolved oxygen content in S2 was the lowest. Dissolved oxygen content will affect biodiversity, community structure, and the circulation of essential nutrients [37–39]. Therefore, the low dissolved oxygen value may have a particular impact on biodiversity and biogeochemical cycles in this section. The pH of the river was weakly alkaline. The permanganate index ($COD_{Mn}$) ranged from 2.6 to 8.3 mg·L$^{-1}$ in the river, with an average content of 6.8 mg·L$^{-1}$. This index was higher at the middle and lower reaches of the river. The value of S7 was the largest, indicating that organic pollutants gradually accumulated in the middle and lower reached after river scouring, resulting in severe organic pollution in the middle and lower reaches [40].

**Table 3.** The $COD_{Mn}$, DO, and pH values of overlying water of the Yitong River.

| Sampling Point | $COD_{Mn}$ | DO | pH |
|---|---|---|---|
| S1 | 2.56 | 5.46 | 6.96 |
| S2 | 3.46 | 4.11 | 7.79 |
| S3 | 5.24 | 7.11 | 7.50 |
| S4 | 7.56 | 9.00 | 8.90 |
| S5 | 7.42 | 8.60 | 9.05 |
| S6 | 5.07 | 6.35 | 8.77 |
| S7 | 8.31 | 8.65 | 7.83 |
| S8 | 7.78 | 5.16 | 7.52 |

3.1.2. Analysis of Water Quality Characteristics of Interstitial Water

The concentrations of $NH_4^+$-N, TDN, and TDP in the interstitial water at each point are shown in Figure 3. It can be seen from Figure 3 that the $NH_4^+$-N concentration was between 0.31 and 16.1 mg·L$^{-1}$, and the $NH_4^+$-N concentration in the interstitial water at each sampling point was higher than that in the overlying water, and the concentration difference was enormous. Therefore, it was inferred that the endogenous $NH_4^+$-N pollutants had a diffusion trend to the overlying water. The $NH_4^+$-N concentration at point S3 was the highest. During sampling, it was found that the water level at this point was high, there was a large amount of sediment accumulation and domestic waste. Because the water body used to hold a large amount of domestic sewage and waste for a long time, resulting in

nitrogen pollutants in the sediments, the content of nitrogen pollutants in the sediment increased, which posed a potential threat to the quality of the Yitong River.

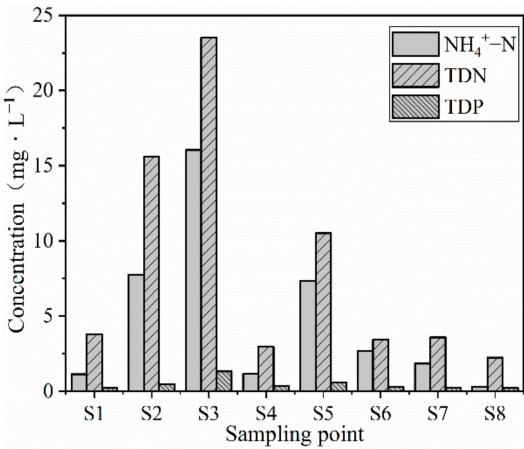

**Figure 3.** Nutrient concentration of interstitial water at each sampling point of the Yitong River.

The average concentration of TDN was 8.2 mg·$L^{-1}$, and the variation range of concentration was 2.2–23.5 mg·$L^{-1}$. The variation trend was the same as $NH_4^+$-N, and the maximum value still appeared at point S3. Except for S2 and S6 sampling points, there was a slight difference in TDN concentration between overlying water and interstitial water. TDN and $NH_4^+$-N nutrient concentrations in the interstitial water at the S2 point were more than 50 times that of the overlying water. Although this point was far away from rural and urban areas, the water depth was shallow, and the water grass grew luxuriantly, resulting in low dissolved oxygen content. Dissolved oxygen content had a significant impact on the release of nitrogen nutrients in sediments. Nitrogen flux reached the maximum under anoxic conditions [41]. Therefore, it was easy to release endogenous nitrogen pollutants. Site S6 was located in Beihu Park, Changchun City. Due to the small impact of domestic sewage, the quality of overlying water was better, but its sediments were affected by reservoir discharge, and some nitrogen and phosphorus pollutants in the middle reaches were washed here. The concentration of nitrogen and phosphorus nutrients in the interstitial water at this point increased, increasing the risk of endogenous nitrogen and phosphorus pollutants released to the water body.

The average concentration of TDP was 0.47 mg·$L^{-1}$, and the concentration variation range was 0.23–1.32 mg·$L^{-1}$. Although it was no different from the phosphorus content in the overlying water body, there was still a risk of phosphorus pollutant release to the overlying water body. Generally speaking, the concentration of nitrogen and phosphorus nutrients in the interstitial water in the urban section was high, but the content of nitrogen pollutants was much higher than that of phosphorus pollutants. The endogenous nitrogen pollution was severe, which would readily cause the migration of endogenous nitrogen pollutants to the overlying water. The concentration of nitrogen and phosphorus nutrients at S7 and S8 points downstream was small. It was found that the water flow speed at this point was fast, the sediment was washed away by the water body, and the surface sediments mainly were sand and gravel, which led to the low concentration of nitrogen and phosphorus nutrients in the interstitial water [42].

### 3.2. Comprehensive Pollution Index Evaluation of Water Quality

In order to determine the pollution status of the overlying water body and the interstitial water body of the Yitong River, the comprehensive pollution index method was used to expansively evaluate the water quality of eight points, as shown in Figure 4. As mentioned above, between the two dotted lines were the heavy pollution level (comprehensive pollution index range $1.0 < p \leq 2.0$), and above the uppermost dotted line was the severe pollution level (comprehensive pollution index range $p > 2.0$).

The comprehensive pollution index value of overlying water quality is shown in Figure 4a. The overlying water's comprehensive pollution index evaluation results at the eight points were between 0.57 and 2.9, showing that the upstream water quality was good and only slightly polluted. The urban section of the middle reaches was severely polluted, and the comprehensive results of S3 and S5 were high due to the aggravation of water pollution caused by excessive discharge of domestic sewage near residential areas. The water bodies at the downstream points were heavily polluted, and the inflow of many pollutants in the middle reaches was the main reason for the poor water quality. Figure 4b shows that interstitial water quality's comprehensive pollution index results were between 1.23 and 15.39. The water quality at the S8 point was heavily polluted, but the water bodies at other points were likewise severely polluted. The interstitial water pollution in the urban section was severe, and the highest total score was still at S3. The comprehensive pollution index evaluation results revealed the pollution degree of overlying water and interstitial water. It indicated that the pollution degree of interstitial water was much higher than that of overlying water. The internal pollution at S3 point was severe, which needed to be treated to prevent secondary pollution to the overlying water body.

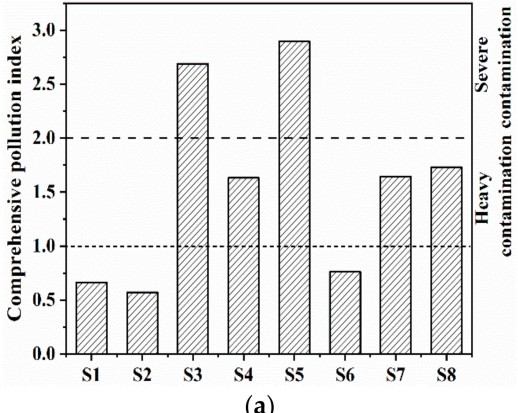 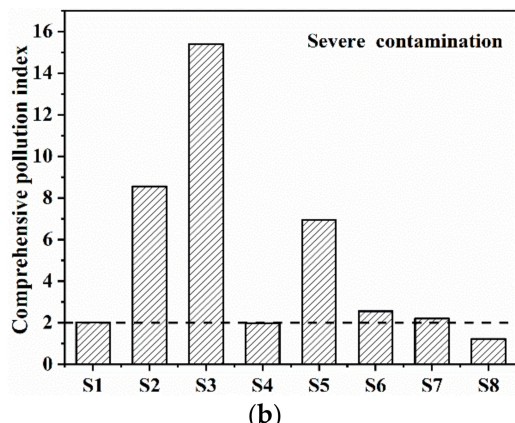

(**a**)  (**b**)

**Figure 4.** Evaluation of comprehensive pollution index of the Yitong River. (**a**) Overlying water, (**b**) Interstitial water.

### 3.3. Correlation Analysis of Nitrogen and Phosphorus between Sediment and Water

In order to explore the migration and transformation process of nitrogen and phosphorus nutrients at the sediment–water interface, Pearson correlation analysis was carried out on the correlation between nitrogen and phosphorus in overlying water, interstitial water, and sediments of the Yitong River with the help of SPSS software (20.0). The results are shown in Table 4. The results showed a correlation between nitrogen and phosphorus in overlying water, but the correlation coefficient was small. The correlation coefficients of nitrogen and phosphorus in interstitial water and sediment were 0.908 and 0.859, respectively, revealing the homology of nitrogen and phosphorus. The correlation coefficients between total phosphorus in overlying water and total nitrogen and total phosphorus in interstitial water were 0.826 and 0.904, respectively, indicating that nitrogen and phosphorus pollutants in interstitial water had a significant impact on the content of phosphorus in the overlying water. There was no significant correlation between nitrogen and phosphorus in sediment and overlying water, but there was a significant correlation between nitrogen and phosphorus in sediment and phosphorus in interstitial water. In terms of nitrogen and phosphorus content, there was a specific correlation between nitrogen and phosphorus contents in overlying water and sediments. However, the correlation coefficient was not high as the sediment–water interface exchange between substances was affected by the concentration gradient and the wind and waves, exogenous input, aquatic biological activities, specific surface area, and sediment particle size [43,44]. Therefore, even if there was a significant concentration difference between nitrogen and phosphorus in the overlying

water and sediments, the nutrients in the sediments did not necessarily affect the overlying water body.

**Table 4.** Correlation analysis of nitrogen and phosphorus in overlying water, interstitial water, and sediment of the Yitong River.

| | Index | Overling Water TN | TP | Interstitial Water TN | TP | Sediment TN | TP |
|---|---|---|---|---|---|---|---|
| Overling water | TN | 1 | 0.658 | 0.198 | 0.381 | 0.222 | 0.265 |
| | TP | | 1 | 0.826 * | 0.904 ** | 0.508 | 0.431 |
| Interstitial water | TN | | | 1 | 0.908 ** | 0.437 | 0.433 |
| | TP | | | | 1 | 0.691 | 0.624 |
| Sediment | TN | | | | | 1 | 0.859 ** |
| | TP | | | | | | 1 |

* Indicates a significant correlation at the 0.05 level, ** indicates a significant correlation at the 0.01 level.

*3.4. Analysis of Nitrogen and Phosphorus Nutrient Exchange Flux at the Sediment–Water Interface*

3.4.1. Diffusion Flux of Nitrogen and Phosphorus Nutrients at the Sediment–Water Interface

In order to explore the mutual diffusion relationship of nitrogen and phosphorus nutrients between the sediment and water interface of the Yitong River and determine whether the sediment was the "source" or "sink" of nitrogen and phosphorus nutrients, this section evaluates the diffusion flux of nitrogen and phosphorus nutrients at the sediment–water interface through Fick's first law. The results are shown in Tables 5–7. When the diffusion flux was positive, it meant that nutrients diffused from sediments to water. Moreover, when it was negative, it meant that nutrients diffused from water to sediments—the greater the diffusion flux, the faster the diffusion speed.

$NH_4^+$-N diffusion flux values are shown in Table 5, ranging from 0.03 to 6.52 mg·$(m^2·d)^{-1}$, and were all positive, indicating that $NH_4^+$-N at each point diffused from sediment to water body. Thus, the sediment was the "source" of ammonia nitrogen nutrients in the water body. Among them, the diffusion rate at point S3 was the largest, the $NH_4^+$-N concentration gradient at the sediment–water interface was the largest, and the $NH_4^+$-N pollutants in the sediments tend to diffuse into the water.

**Table 5.** Porosity, diffusion coefficient and diffusion flux of $NH_4^+$-N at the sediment–water interface of the Yitong River.

| Index | Sampling Point | $\Phi$/% | $D_0 \times 10^{-6}$/cm$^2$·s$^{-1}$ | $D_S \times 10^{-6}$/cm$^2$·s$^{-1}$ | F/mg $(m^2 \cdot d)^{-1}$ |
|---|---|---|---|---|---|
| $NH_4^+$-N | S1 | 0.538 | 9.80 | 5.274 | 0.49 |
| | S2 | 0.423 | 9.80 | 5.456 | 3.99 |
| | S3 | 0.538 | 9.80 | 5.272 | 6.52 |
| | S4 | 0.490 | 9.80 | 4.797 | 0.43 |
| | S5 | 0.507 | 9.80 | 4.971 | 1.95 |
| | S6 | 0.478 | 9.80 | 4.686 | 1.00 |
| | S7 | 0.416 | 9.80 | 4.078 | 0.47 |
| | S8 | 0.509 | 9.80 | 4.993 | 0.02 |

TDN diffusion flux values are shown in Table 6. There was a considerable difference among sampling points, in which the diffusion flux at S4, S7, and S8 points were negative, which indicated that the sediment served as a sink for nitrogen pollutants in water. Site S4 was located in the urban section, and the nitrogen pollution of the overlying water body was very severe. It was found by on-site sampling that the water flow speed was fast and affected by the opening of the sluice. As a result, there was more sediment loss, and the TDN concentration of sediment interstitial water decreased. Therefore, the nitrogen pollutants at this point diffused from the overlying water body to the sediments. In addition, there was farmland around points S7 and S8. As a result, pesticides and chemical fertilizers flowed into the river through rainwater, resulting in deterioration of water quality, and nitrogen pollutants in the overlying water diffused to interstitial water and accumulated in the sediments. The diffusion flux values of the other five points ranged from 1.67 to 11.66 mg·$(m^2·d)^{-1}$, among which the diffusion rate of TDN at S2 and S3 was the fastest.

According to the results of $NH_4^+$-N and TDN diffusion fluxes, although the diffusion rate of endogenous nitrogen pollutants varied greatly at each point, $NH_4^+$-N and TDN in the sediment at S3 point diffuse fastest to the overlying water body.

**Table 6.** Porosity, diffusion coefficient, and diffusion flux of TDN at the sediment–water interface of the Yitong River.

| Index | Sampling Point | $\Phi$/% | $D_0 \times 10^{-6}$/cm$^2 \cdot$s$^{-1}$ | $D_S \times 10^{-6}$/cm$^2 \cdot$s$^{-1}$ | F/mg$\cdot$(m$^2 \cdot$d)$^{-1}$ |
|---|---|---|---|---|---|
| | S1 | 0.538 | 14.21 | 7.647 | 1.81 |
| | S2 | 0.557 | 14.21 | 7.911 | 6.08 |
| | S3 | 0.538 | 14.21 | 7.645 | 11.59 |
| | S4 | 0.490 | 14.21 | 6.956 | −0.65 |
| TDN | S5 | 0.507 | 14.21 | 7.207 | 1.70 |
| | S6 | 0.478 | 14.21 | 6.795 | 1.67 |
| | S7 | 0.416 | 14.21 | 5.913 | −0.40 |
| | S8 | 0.509 | 14.21 | 7.240 | −1.57 |

The diffusion flux values of TDP at the eight sampling points are shown in Table 7. The diffusion flux at S8 point was negative, which may be mainly due to the high content of phosphorus pollutants in the water body due to the direct discharge of domestic sewage into the river, which led to the diffusion of phosphorus pollutants from the water body to the sediment. The TDP diffusion flux at other points ranged from 0.01 to 0.22 mg$\cdot$(m$^2 \cdot$d)$^{-1}$, with an average of 0.07 mg$\cdot$(m$^2 \cdot$d)$^{-1}$, and the highest diffusion rate still appeared at S3.

**Table 7.** Porosity, diffusion coefficient and diffusion flux of TDP at the sediment–water interface of the Yitong River.

| Index | Sampling Point | $\Phi$/% | $D_0 \times 10^{-6}$/cm$^2 \cdot$s$^{-1}$ | $D_S \times 10^{-6}$/cm$^2 \cdot$s$^{-1}$ | F/mg (m$^2 \cdot$d)$^{-1}$ |
|---|---|---|---|---|---|
| | S1 | 0.538 | 6.12 | 3.294 | 0.06 |
| | S2 | 0.557 | 6.12 | 3.407 | 0.11 |
| | S3 | 0.538 | 6.12 | 3.293 | 0.22 |
| | S4 | 0.490 | 6.12 | 2.996 | 0.02 |
| TDP | S5 | 0.507 | 6.12 | 3.104 | 0.02 |
| | S6 | 0.478 | 6.12 | 2.927 | 0.04 |
| | S7 | 0.416 | 6.12 | 2.547 | 0.01 |
| | S8 | 0.509 | 6.12 | 3.118 | −0.05 |

3.4.2. Effect of Nitrogen and Phosphorus Nutrients Diffusion in Sediments on Water Quality

In order to further clarify the impact of the diffusion of nitrogen and phosphorus nutrients in the Yitong River sediments on the overlying water body, the contribution rate of endogenous pollutants to the overlying water was estimated using formula (6) by combining the index of diffusion flux, overlying water concentration, water depth, and water retention time. The calculation results are shown in Tables 8–10. The contribution rate of sediment $NH_4^+$-N diffusion on the water body is shown in Table 8, ranging from 6.17% to 78.83%. The contribution rate of endogenous $NH_4^+$-N on overlying water at S2 and S3 points was similar. For the S2 point, the concentration difference was significant. However, due to the shallowest water depth and substantial impact of wind–wave disturbance on the release of endogenous pollutants, this point has a maximum contribution on overlying water. Although the diffusion flux at the S3 point was the largest, the water depth was deep, attenuating the contribution of sediment $NH_4^+$-N release on the overlying water body. The sediment $NH_4^+$-N diffusion at S8 had the most negligible contribution to the overlying water body, mainly because the concentration gradient formed by the overlying water and interstitial water was slight. The diffusion flux value was only 0.02 mg$\cdot$(m$^2 \cdot$d)$^{-1}$, resulting in the most negligible contribution to the overlying water body [45].

As shown in Table 9, the contribution rate range of endogenous TDN diffusion on the overlying water body was −21.3 to 79.67%. Among them, TDN diffusion in the water at the S8 point had a significant contribution on the sediment, mainly because the overlying water at this point was seriously polluted by nitrogen and had a significant concentration difference with the interstitial water, resulting in the accumulation of nitrogen pollutants in the water body into the sediment. On the other hand, TDN diffusion at S2 still had

the most significant contribution rate on the water body, with a contribution rate range of 79.67%. Thus, according to the table's data, the water body's depth directly determined the contribution rate of endogenous TDN diffusion on the overlying water body.

**Table 8.** Contribution rate of $NH_4^+$-N nutrient diffusion at the sediment–water interface to the water body.

| Index | Sampling Point | $F/mg \cdot (m^2 \cdot d)^{-1}$ | H/m | $c/mg \cdot L^{-1}$ | a/% |
|---|---|---|---|---|---|
| | S1 | 0.49 | 1.55 | 0.15 | 19.16 |
| | S2 | 3.99 | 0.61 | 0.13 | 78.83 |
| | S3 | 6.52 | 1.83 | 2.73 | 78.34 |
| $NH_4^+$-N | S4 | 0.43 | 1.95 | 0.11 | 17.96 |
| | S5 | 1.95 | 2.83 | 2.87 | 14.40 |
| | S6 | 1.00 | 1.12 | 0.11 | 70.45 |
| | S7 | 0.47 | 0.87 | 0.23 | 20.95 |
| | S8 | 0.02 | 0.94 | 0.26 | 6.17 |

Note: in the table, F is the diffusion flux, H is the water depth, c is the concentration of overlying water, and a is the contribution rate of the diffusion of endogenous nutrients to the overlying water body.

**Table 9.** Contribution rate of TDN nutrient diffusion at the sediment–water interface to the water body.

| Index | Sampling Point | $F/mg \cdot (m^2 \cdot d)^{-1}$ | H/m | $c/mg \cdot L^{-1}$ | a/% |
|---|---|---|---|---|---|
| | S1 | 1.81 | 1.55 | 1.25 | 56.24 |
| | S2 | 6.08 | 0.61 | 0.28 | 79.67 |
| | S3 | 11.59 | 1.83 | 7.16 | 53.07 |
| TDN | S4 | −0.65 | 1.95 | 4.06 | −4.89 |
| | S5 | 1.70 | 2.83 | 7.82 | 4.60 |
| | S6 | 1.67 | 1.12 | 0.48 | 62.50 |
| | S7 | −0.40 | 0.87 | 4.52 | −6.07 |
| | S8 | −1.57 | 0.94 | 4.71 | −21.30 |

Note: in the table, F is the diffusion flux, H is the water depth, c is the concentration of overlying water, and a is the contribution rate of the diffusion of endogenous nutrients to the overlying water body.

The contribution rate of sediment TDP at each point on water quality is shown in Table 10. Although the diffusion flux at the S2 point was not the largest, the pollution to the overlying water body was the most serious, with a contribution rate range of 95.64%. On the other hand, the value of the endogenous TDP diffusion at point S1 on the water body had reached 86.9%. At this point, the diffusion flux at the sediment–water interface was small, but there was a significant concentration difference between them, resulting in an excellent contribution rate of sediments on the water quality of the overlying water body.

**Table 10.** Contribution rate of TDP nutrient diffusion at the sediment–water interface to the water body.

| Index | Sampling Point | $F/mg \cdot (m^2 \cdot d)^{-1}$ | H/m | $c/mg \cdot L^{-1}$ | a/% |
|---|---|---|---|---|---|
| | S1 | 0.06 | 1.55 | 0.03 | 86.90 |
| | S2 | 0.11 | 0.61 | 0.12 | 95.64 |
| | S3 | 0.22 | 1.83 | 0.59 | 12.39 |
| TDP | S4 | 0.02 | 1.95 | 0.27 | 2.66 |
| | S5 | 0.02 | 2.83 | 0.51 | 1.04 |
| | S6 | 0.04 | 1.12 | 0.14 | 14.97 |
| | S7 | 0.01 | 0.87 | 0.19 | 3.19 |
| | S8 | −0.05 | 0.94 | 0.42 | −7.95 |

Note: in the table, F is the diffusion flux, H is the water depth, c is the concentration of overlying water, and a is the contribution rate of the diffusion of endogenous nutrients to the overlying water body.

## 4. Conclusions

The content of nitrogen and phosphorus in overlying water in the urban section of the Yitong River exceeded the standard, and the content of nitrogen and phosphorus pollutants in interstitial water was higher. The maximum values of TDN and TDP reached 23.5 mg·$(m^2 \cdot d)^{-1}$ and 0.6 mg·$(m^2 \cdot d)^{-1}$, respectively, which are severely polluted except for the point Kaoshan Bridge (S8). There was a pronounced diffusion gradient between

overlying water and interstitial water, and the nitrogen and phosphorus pollutants in sediments tend to diffuse to the overlying water body. The diffusion fluxes of $NH_4^+$-N in the sediments of the Yitong River were all positive, which were the "source" of ammonia nitrogen pollutants in the water body. The diffusion fluxes ranged from 0.03 to 6.52 mg·$(m^2 \cdot d)^{-1}$. The values of TDN diffusion flux at each point were quite different, and the sediment was both the "source" and "sink" of nitrogen pollutants. Except for the sampling point of the Kaoshan Bridge, TDP was the diffusion of sediment into the water body, the diffusion flux ranged from $-0.05$ to 0.22 mg·$(m^2 \cdot d)^{-1}$, and the diffusion rate was relatively low. Among the three indicators, the content of nitrogen and phosphorus pollutants in the South Third Ring Road (S3) sediments was the highest, and the release rate was the fastest.

By estimating the contribution rate of nitrogen and phosphorus nutrients in the Yitong River sediments to the overlying water body, it was found that the release of endogenous nitrogen and phosphorus pollutants at Xinlicheng Reservoir (S2) had the most significant contribution on water quality. The contribution rate range of endogenous $NH_4^+$-N, TDN, and TDP on the overlying water was 78.83%, 79.67%, and 95.64%, respectively, indicating the release of endogenous nitrogen and phosphorus pollutants at this point was a severe hidden danger affecting the overlying water quality. This study revealed the water quality characteristics of the Yitong River and the exchange law of nitrogen and phosphorus between the sediment–water interface to a certain extent, but it only considered the relationship between the sediment and overlying water, ignoring the influence of interfacial biological interaction on material diffusion. Future research could focus on the nutrient exchange flux at the sediment–water interface of rivers and other surface water and the relationship between the three factors and the key species affecting the diffusion of substances at the interface.

**Author Contributions:** Conceptualization, K.Z. and H.L.; methodology, H.L.; software, H.F.; validation, Q.W.; formal analysis, H.F.; investigation, H.F. and Q.W.; resources, K.Z.; data curation, K.Z.; writing—original draft preparation, K.Z.; writing—review and editing, H.L.; visualization, H.L.; supervision, H.L.; project administration, K.Z.; funding acquisition, K.Z. All authors have read and agreed to the published version of the manuscript.

**Funding:** This work was supported by the Key R&D Program of Department of Science and Technology of Jilin Province (NO. 20210203035SF).

**Institutional Review Board Statement:** Not applicable.

**Informed Consent Statement:** Not applicable.

**Conflicts of Interest:** The authors declare no conflict of interest.

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
