# Peer review of "Determination of Water Quality Characteristics and Nutrient Exchange Flux at the Sediment—Water Interface of the Yitong River in Changchun City, China"

_water, doi:10.3390/w13243555_

Round 1
Reviewer 1 Report
Reviewer Comments
Manuscript Number: Water 2021 -1483279
Authors of this manuscript studied the water quality characteristics of the Yitong River and presented their outcomes in this article. They analysed the release characteristics of nutrients at the interface to determine a mutual diffusion relationship of nutrients between sediments and overlying water.
Authors have included and presented well some important results in the manuscript. However, the authors are requested to respond the following comments and suggestions to improve the quality of the manuscript further to publish in Water.
- Abstract: A well written abstract. However, it does not include sufficient amount of facts on “significance and novelty” of the study and the “methods” applied to conduct the research for the readers to understand study well. Please include a several informative statements to highlight these key facts.
- Introduction; Line 41-43: Please support this statement, “When the water environment is disturbed, the nutrient substances with high content in sediment are released to the overlying water by diffusion, convection and resuspension” by including an appropriate reference
- Introduction: The presented study looks like a “case study” rather than a research useful for a broader applications. Please clarify.
- Introduction: Please try to highlight the novelty of the study and the findings can be applicable or important for similar studies conducted for surface water flows across the World
- Materials and Methods; Table 1: Please include appropriate references or details of analytical standards for the determination methods included in Table 1
- Materials and Methods; Section 2.3.1: Please include some more description on Equation (1) to understand the fundamentals of surface water polluting mechanism depicted from this expression better. How many pollution indices used in this study? Please clarify whether all the indicators shown in Table 1 are considered as pollution indices in this study. As it is not very clear, please highlight this fact in the manuscript
- Materials and Methods; Section 2.3.1-3: Please include input data table/s to show the key parameters used to solve Equations (1-6) together with sources of references and specific values of C0 (Class III standard limits) values considered in this study
- Materials and Methods: Please include the calculation/estimation methods and software used to solve the Equations (1) through (6)
- Results and discussion; Line 173-176: It stated “Among them, the concentration of NH4+-N in the two adjacent residential areas of urban section S3 and S5 was high, while the NH4+-N values of other points were within the standard limit of Class II water body of surface water” in the manuscript. What is the standard limit of Class II water body of surface water discharges from urban sections? Please go through the manuscript again try to include all necessary information similar to this
- Results and discussion; Line 181-183: It concluded saying that “This may be due to the severe nitrogen pollution caused by the significant content of nitrogen pollutants in the middle reach of the river flowing downstream”. Please justify/validate this statement or the outcomes obtained from this study using the findings from similar studies conducted by others previously
- Results and discussion; Line 187: It stated “Generally speaking, the water quality….” in the manuscript. Please try to stick to academic writing style only
- Results and discussion; Line 198-205: Please depict the CODMn results obtained for all Sections in Figure 1 or in a separate Table
- Results and discussion; Sections 3.1 to 3.4: The concluding statements included in each Section look vague, hence please rephrase to make them specific concluding statements and support the predictions/assumptions using appropriate literature information
- Results and discussion; Line 367: It stated that “as shown in Table 4-2, the influence range of endogenous TDN diffusion on…”. It should be endogenous TDN diffusion “rate”. Please use the accurate term and the symbols used for such parameters when describing in text
- Results and discussion; Data Tables: Please check the Table Numbering and correct them to match with the standard method stipulated in the Water author guide
- Results and discussion; Table 4-1 to Table 4-3; Page 11 of 13: Please include the definitions of all the parameters shown in these data Tables and describe the estimating methods to be consistent with other parameter estimations depicted in Materials and Methods Section
- Conclusions: Please try to give a message to the future researchers who would conduct similar studies on water quality characteristics and investigating nutrient exchange fluxes at the sediment-water interfaces of rivers and other surface water bodies in the World
- References: Most of the references used in this article are quite old. Many recent scholarly articles are available and can be used to support the facts included in this manuscript. Please include a set of recent references
Author Response
Reviewer #1:
- Abstract: A well written abstract. However, it does not include sufficient amount of facts on “significance and novelty” of the study and the “methods” applied to conduct the research for the readers to understand study well. Please include a several informative statements to highlight these key facts.
Thank you for your comments. The Abstract section has been modified. See lines 10-13 and 28-31 for details.
- Introduction; Line 41-43: Please support this statement, “When the water environment is disturbed, the nutrient substances with high content in sediment are released to the overlying water by diffusion, convection and resuspension” by including an appropriate reference
Thank you for your comment. Reference has been added, i.e., references 7-9 in the list.
- Introduction: The presented study looks like a “case study” rather than a research useful for a broader applications. Please clarify.
Thank you for your comment. We have clarified this issue. Kindly see line 72.
- Introduction: Please try to highlight the novelty of the study and the findings can be applicable or important for similar studies conducted for surface water flows across the World
Thank you for your comments. We have revised the introduction section. See lines 76-82.
- Materials and Methods; Table 1: Please include appropriate references or details of analytical standards for the determination methods included in Table 1
Thanks for your comment, we have added references to the content of Table 1, as detailed in the title of Table 1.
- Materials and Methods; Section 2.3.1: Please include some more description on Equation (1) to understand the fundamentals of surface water polluting mechanism depicted from this expression better. How many pollution indices used in this study? Please clarify whether all the indicators
Thank you for your comments. We have described Formula 1 in detail. We selected the most representative 7 parameters in this experiment instead of all parameters, as shown in lines 128-130.
- shown in Table 1 are considered as pollution indices in this study. As it is not very clear, please highlight this fact in the manuscript
Thank you for your comments. We have made changes to the text. See lines 122-124.
- Materials and Methods; Section 2.3.1-3: Please include input data table/s to show the key parameters used to solve Equations (1-6) together with sources of references and specific values of C0(Class III standard limits) values considered in this study
Thank you for your comments. Table 3 has been added to the results and discussion section. The parameter data used in the present formula (1-6) are presented in each chapter of the results and discussion section. The standard limits of surface water in this study have been added to Table 2, and reference [31] has been added at the title of Table 2.
- Materials and Methods: Please include the calculation/estimation methods and software used to solve the Equations (1) through (6)
Thank you for your comment. We have amended to include 2.3.4; see lines 183-188.
- Results and discussion; Line 173-176: It stated “Among them, the concentration of NH4+-N in the two adjacent residential areas of urban section S3 and S5 was high, while the NH4+-N values of other points were within the standard limit of Class II water body of surface water” in the manuscript. What is the standard limit of Class II water body of surface water discharges from urban sections? Please go through the manuscript again try to include all necessary information similar to this
Thank you for your comment. We have modified it. See lines 197 and 210.
- Results and discussion; Line 181-183: It concluded saying that “This may be due to the severe nitrogen pollution caused by the significant content of nitrogen pollutants in the middle reach of the river flowing downstream”. Please justify/validate this statement or the outcomes obtained from this study using the findings from similar studies conducted by others previously
Thank you for your comment. We have revised to include reference 42.
- Results and discussion; Line 187: It stated “Generally speaking, the water quality….” in the manuscript. Please try to stick to academic writing style only
Thank you for your comment. We have revised it. See line 208.
- Results and discussion; Line 198-205: Please depict the CODMnresults obtained for all Sections in Figure 1 or in a separate Table
Thanks for your comment; we have revised the results and discussion section to include Table 3.
- Results and discussion; Sections 3.1 to 3.4: The concluding statements included in each section look vague, hence please rephrase to make them specific concluding statements and support the predictions/assumptions using appropriate literature information
Thank you for your comment, we have revised the results and discussion section, see lines 272-273, along with the addition of literature 40, 42, 45.
- Results and discussion; Line 367: It stated that “as shown in Table 4-2, the influence range of endogenous TDN diffusion on…”. It should be endogenous TDN diffusion “rate”. Please use the accurate term and the symbols used for such parameters when describing in text
Thank you for your comment. According to your suggestion, the results and discussion have been modified, see lines 376-391, 395-403, and 408-414; line 439 of the conclusion section has been changed.
- Results and discussion; Data Tables: Please check the Table Numbering and correct them to match with the standard method stipulated in the Water author guide
Thank you for your comments. We have revised tables 4-1, 4-2 and 4-3 into tables 5, 6 and 7; and replaced tables 5-1, 5-2 and 5-3 with tables 8, 9 and 10.
- Results and discussion; Table 4-1 to Table 4-3; Page 11 of 13: Please include the definitions of all the parameters shown in these data Tables and describe the estimating methods to be consistent with other parameter estimations depicted in Materials and Methods Section
Thank you for your comments. We have revised the results and discussion and annotated them below table 8, table 9 and table 10. The specific description of the estimation method of formula (6) is added; see lines 376-379.
- Conclusions: Please try to give a message to the future researchers who would conduct similar studies on water quality characteristics and investigating nutrient exchange fluxes at the sediment-water interfaces of rivers and other surface water bodies in the World
Thank you for your comments. We have revised the conclusion section; see lines 443-449.
- References: Most of the references used in this article are quite old. Many recent scholarly articles are available and can be used to support the facts included in this manuscript. Please include a set of recent references
Thanks for your comment, we have added references from recent years to the list of 1,3,4,6,13,14,16,31,40,42,45.

Reviewer 2 Report
This paper discusses the determination possiblities of the nutrient exchange flux at the sediment-water interface of freshwater bodies in order to prevent and control water eutrophication at the Yitong River in Changchun, China. The Abstract is well done just must add methodology sentences. In addition, the introduction showed the need for this study, some references are missing. The results are well presented and supported by the conclusion form.
Here attached some remarks to correct!!

Author Response
Reviewer #2:
- please change the title to “Determination of water quality…city China”
Thank you for your comments. We have revised the title accordingly.
- Abstract: Line 12: Where exactly(country)?
Thanks for your comment; we have changed the abstract section, see line 13.
- Abstract: Line 14, Please add one sentence talking about the followed methodology in this study.
Thank you for your comments. We have modified the abstract section accordingly; see lines 10-13.
- Abstract: Line16, 26: How can I know what does it mean this one?
Thank you for your comments. We have revised the Abstract; see lines 15, 25 and 26.
- Introduction: Line 36: Please provide references for this sentence.
Thank you for your comments. We have revised the quotation and added reference 1.
- Introduction: Line 43: References.
Thank you for your comments. We have revised the introduction and added references 7-9.
- Remove the word “Legend” from the mape,improve the resolution of the map.add more indication of geographical situation.(country,region,province and city).
Thank you for your comments. We have modified the map, as shown in Figure 1.
- pH: power of hydrogen
Thank you for your comments. We have revised the introduction; see lines 114-115.
